# Effect of Parasitic Native Plant *Cuscuta australis* on Growth and Competitive Ability of Two Invasive Xanthium Plants

**DOI:** 10.3390/biology13010023

**Published:** 2023-12-31

**Authors:** Jianxiao He, Yongkang Xiao, Amanula Yimingniyazi

**Affiliations:** 1Key Laboratory of Grassland Resources and Ecology of the Ministry of Education in Western Arid Desert Region, College of Grassland Sciences, Xinjiang Agricultural University, Urumqi 830052, China; xjau1952@163.com (J.H.); xiao_yk2023@163.com (Y.X.); 2Xinjiang Key Laboratory for Ecological Adaptation and Evolution of Extreme Environment Biology, College of Life Sciences, Xinjiang Agricultural University, Urumqi 830052, China; 3Institute of Plant Protection, Xinjiang Academy of Agricultural Sciences, Key Laboratory of Integrated Pest Management on Crops in Northwestern Oasis, Ministry of Agriculture, Xinjiang Key Laboratory of Agricultural Biosafety, Urumqi 830091, China

**Keywords:** competitiveness, *Cuscuta australis*, *Xanthium spinosum*, *Xanthium italicum*, relative competitive intensity (RCI)

## Abstract

**Simple Summary:**

The impact of parasitic plants on invasive plants varies depending on the parasitism time and species. *Cuscuta australis* is a common parasitic plant, while *Xanthium spinosum* and *Xanthium italicum* are invasive plants. The impact of parasitism on the competitive ability of invasive plants has been rarely reported. Therefore, in this study, we aim to investigate the impact of southern dodder parasitism on the competitive ability of *X. spinosum* and *X. italicum*. At the same time, we found that parasitism of southern dodder increased the competitive ability of *X. spinosum* and weakened the competitive ability of *X. italicum*. It is worth noting that invasive plants also provide a medium for the spread of southern dodder. Based on the research results, we suggest trying to prevent the same domain distribution of the two as much as possible to reduce the harm to local plants and crops.

**Abstract:**

The competitive ability of invasive plants is a key factor in their successful invasion, and research on this ability of invasive plants can provide a theoretical basis for the prevention and control of invasive plants. This study used *Cuscuta australis*, *Xanthium spinosum*, and *Xanthium italicum* as research materials and conducted outdoor controlled pot experiments to compare and study the changes in the biomass, competitiveness, and growth cycle of *X. spinosum* and *X. italicum* parasitized by *C. australis* at different growth stages. The results showed that (1) parasitism by *C. australis* increased the biomass of *X. spinosum* and decreased that of *X. italicum*, but under parasitism, the root cap ratio of *X. spinosum* and *X. italicum* increased, and the fruit biomass ratio decreased, indicating that *X. spinosum* and *X. italicum* reduced the energy input for reproduction and increased the energy input for nutrient growth to resist the impact of *C. australis* parasitism; (2) the relative competitive intensity calculated based on the total biomass of a single plant showed a negative value for *X. spinosum* during parasitism at the flowering and fruit stages, indicating an increase in competitive ability, and *X. italicum* showed a positive value during parasitism at the seedling and flowering stages, indicating a decrease in competitive ability; and (3) the parasitism of *C. australis* significantly shortened the fruit stage of *X. spinosum* and *X. italicum*, leading to a significant advance in their flowering, fruiting, and fruit ripening times. Simultaneously, it significantly reduced the morphological indicators of biomass, plant height, and crown width. Thus, *C. australis* parasitism has a certain inhibitory effect on the competitive ability of some invasive plants and can shorten their growth cycle, the latter of which has an important impact on their reproduction and diffusion.

## 1. Introduction

Plant competition refers to the interrelationships between plants when the required environmental resources or space are relatively insufficient [1]. Competitiveness is a key factor that determines whether exotic plants successfully invade new environments [2]. When strongly competitive exotic plants compete for resources with invasive species, they can successfully exclude local plants, greatly reducing the biodiversity of the invasive area and disrupting the ecological balance [3]. Therefore, studying the competitiveness of exotic plants has become a hot topic in invasion ecology.

Parasitic plants, as local natural enemies, play an important regulatory role in alien plant populations and a decisive role in the invasion process of alien plants [4], and are important natural enemies that absorb water and nutrients from their host plants (e.g., invasive plants such as *Solidago canadensi* L. and *Alternanthera philoxeroides* (Mart.) Griseb.) through suckers. Simultaneously, because the photosynthesis of the host plant is affected, its growth inhibition exceeds the accumulation of its biomass, affecting its growth, development, and competitiveness [5,6].

Hypotheses related to exotic plants and their natural enemies include the enemy release hypothesis, biotic resistance from enemies hypothesis, evolution of increased competitive ability hypothesis, evolutionary reduced competitive ability hypothesis, biotic resistance from competitors hypothesis, and the new association hypothesis of competitive objects. Enemy release and biotic resistance from enemies suggest that when invading plants enter a new environment, local or exotic natural enemies may prevent the invasion of exotic plants through predation or parasitism or alleviate the invasion pressure by inhibiting or delaying settlement [7,8,9]. The evolution of increased competitive ability suggests that exotic plants in their native areas need to invest certain resources for defense owing to the influence of natural enemies, which can affect their growth and reproduction. In invasive areas, due to the lack of natural enemies, resources originally used for defense can be used for growth and reproduction, improving their competitiveness [10,11,12]. The evolutionarily reduced competitive ability suggests that if there is less competition in invasive areas and competition involves adaptive cost characteristics, invasive species may evolve in a direction that has adverse effects on them, reducing intraspecies interactions [13,14]. The biotic resistance of competitors suggests that in new environments, exotic plants compete with local or other exotic plants to prevent their invasion by inhibiting their settlement, domestication, and persistence through competition for nutrients, water, light, and other resources [15,16]. These new associations suggest that invasive species form new relationships with other species in the community, and the impact of this relationship on alien plants usually manifests as promoting or preventing the successful invasion of new habitats by alien plants.

According to field investigations in the suburbs of Urumqi, Xinjiang, *Cuscuta australis* can effectively parasitize two invasive plants in the composite family, *Xanthium spinosum* and *Xanthium italicum*. *X. spinosum* is an invasive weed native to South America [17]. This species has a strong growth and reproduction ability, high seed yield, and numerous dispersal media, and can quickly occupy a large area, inhibiting the growth and reproduction of local plants and crops [18]. The entire plant has slight toxicity and many sharp yellow thorns in its long petiole; thus, cattle and sheep do not feed on this plant, which directly or indirectly affects the development of agriculture and animal husbandry in the invaded area [19]. *X. italicum*, an annual invasive weed, is native to North America [20]. This species has a well-developed root system, strong ecological adaptability, large growth capacity, high seed-setting rate, wide seed transmission pathways, and seeds with thorns, which seriously affect the local ecological environment, animal husbandry, and agriculture [21]. *C. australis* is a parasitic plant of *Cuscuta* spp. in the family Convolvulaceae [22] and is recognized as a harmful weed in agriculture and forestry [23,24].

With the increasing severity of global biological invasions, the use of local parasitic plants to prevent and control exotic plants, known as “grass control”, has become a research hotspot. Moreover, studying the relationship between *C. australis* and two invasive species of *Xanthium* would provide a theoretical basis for the biological control of these two invasive species. Studies have shown that plants belonging to the genus *Cuscuta* effectively inhibit the growth of invasive plants and restore local communities. For example, parasitism by *C. australis* significantly reduces the net assimilation rate (NAR) and relative growth rate (RGR) of *Bidens pilosa* L. [25], and parasitism by *C. australis* of *A. philoxeroides* can significantly increase its root-to-shoot ratio and community diversity, promoting the recovery of local communities [26]. 

Based on field observations and the literature, we hypothesized that parasitism by *C. australis* would affect the biomass allocation and competitiveness of *X. spinosum* and *X. italicum*. However, no relevant reports have investigated the impact of the parasitism of *C. australis* on the competitive ability of the two species of *Xanthium*. Therefore, in this study, we focus on the changes in the biomass, allocation, and competitiveness of *X. spinosum*, *X. italicum*, and *C. australis* during their parasitism, seedling parasitism, flowering parasitism, and fruiting parasitism. 

The aim was to understand the parasitic relationship between *C. australis*, *X. spinosum*, and *X. italicum*; evaluate whether parasitic plants can become effective biological control agents to provide a theoretical basis for the biological control of invasive plants; and answer three scientific questions, namely: (1) How does the biomass of *X. spinosum* and *X. italicum* parasitizing *C. australis* in southern China change? (to verify the growth-defense trade-off and resource availability hypothesis); (2) How does the parasitism of *C. australis* affect the relative intensity of competition between *X. spinosum* and *X. italicum*? (to verify the evolution of the increased competitive ability and the evolutionary reduced competitive ability hypothesis); and (3) What is the effect of parasitism of *C. australis* on the growth cycles of *X. spinosum* and *X. italicum*? 

## 2. Materials and Methods

Seed collection: *C. australis*, *X. spinosum*, and *X. italicum* were the experimental materials collected from dry plants from March to April 2021. Seeds of *C. australis* were collected from plants near the Sanping Farm in Urumqi. Seeds of *X. spinosum* and *X. italicum* were collected from dry plants along the forest belt and roads of Sanping Farm in Urumqi. (Table 1) For seed collection, 50 plants were randomly selected from each natural population, and from three to five fruits were collected from each plant. Seeds collected from all *X. spinosum* plants were mixed. Similarly, the seeds collected from all selected *X. italicum* plants were mixed. The normally developing seeds were placed in labeled envelopes and stored in a refrigerator at 4 °C for the subsequent potted experiments.

### 2.1. Study Area

The experiment was conducted at the Sanping Teaching and Internship Base of Xinjiang Agricultural University. The geographical coordinates were 43°56′ N, 87°20′ E, and the altitude was 790 m. The site was on an alluvial plain on the southern edge of the Junggar Basin, which has a typical continental temperate desert climate: hot and dry in the summer and cold in the winter. The average annual temperature was 7.2 °C, the average annual precipitation was 194.3 mm, and the soil was desert clay.

### 2.2. Experimental Design 

In May 2021, the three plant seeds were placed in degradation bags containing built-in labels. The bags were buried in a storage box containing sterilized sand moistened with distilled water. The seeds were then subjected to low-temperature stratification treatment at 4 °C for 2 weeks to improve the germination rate of southern dodder and two types of xanthium seeds [27].

(1) Competitive ability: The samples were divided into two groups, single species and mixed species, based on Feng [27]. One plant was placed in each flowerpot for a single species, and two plants of the same species were placed in each flowerpot for the mixed species (one plant was not treated, and the other plant was treated at different stages of parasitism). Each plant was treated using four treatments: not parasitism (control group), seedling parasitism, flowering parasitism, and fruiting parasitism. Each treatment had 10 replicates; thus, there were 160 flowerpots. For the parasitic treatment group in the seedling stage, the seeds of *C. australis* were sowed at a distance of 5 cm from the target plant seeds 5 d after sowing to ensure that the target plant emerged earlier than *C. australis*. After the emergence of *C. australis*, one successful parasitic plant of *C. australis* was retained, and the excess *C. australis* was removed. For the flowering parasitism treatment group, the seeds of *C. australis* were sowed at a distance of 5 cm from the target plant when the target plant began to bud, ensuring that *C. australis* could successfully parasitize when the target plant reached flowering. After *C. australis* emerged, one successfully parasitized *C. australis* plant was retained, and the excess plants were removed. During the fruit stage, when the flowers of the target plant began to shed, the *C. australis* was sown 5 cm from the target plant. After *C. australis* emerged, a successfully parasitized *C. australis* plant was retained, and additional plants were removed. Mixed planting involved the parasitic treatment of only one plant in the pot.

(2) Phenology (Growth time): The plants were divided into treatment and control groups by using a single planting method (Figure 1). The treatment group received the parasitic treatment with *C. australis*, and the control group did not receive parasitical treatment with *C. australis* (Figure 1). From three to five seeds of the three types of plants were sown into flowerpots, and 5 d after sowing, from three to five seeds of *C. australis* were sown into flowerpots in the treatment group, 5 cm from the plants. After the target plant and *C. australis* emerged, one target plant with consistent growth in each flowerpot and one successfully parasitized *C. australis* were retained, and excess plants were removed.

Starting from the emergence of *X. spinosum* and *X. italicum*, the emergence time, bud emergence time, flowering time, fruiting time, and fruit ripening time of the *X. spinosum* and *X. italicum* treatment and control groups were recorded.

(3) Collection and measurement: After the fruits of *X. spinosum* and *X. italicum* matured, each planting plant was separated into the root, stem, leaf, and fruit and placed in a brown paper bag with corresponding labels. After being transported to the laboratory, they were blanched at 105 °C for 20 min and dried at 70 °C to constant weight. The biomass of each component of the three invasive plants was measured using a percentile electronic balance with an accuracy of 0.01 (J-SKY).

(4) Data analysis: Excel was used for preliminary data integration. SPSS 26.0, one-way ANOVA, and Origin 2018 were used for thedata analysis.
Biomass change rate = (*Pmix* − *Pmono*)/*Pmono*
Relative Competitive Intensity (RCI) = (*Pmono* − *Pmix*)/*Pmono*

*Pmono* represents the average value of the plant height, crown width, or biomass for a single species, and *Pmix* represents the corresponding value of each indicator for the mixed species.

## 3. Results

The total and aboveground biomass of *X. spinosum* parasitized during the flowering and fruiting stages of *C*. *australis* increased by 48%, 10%, 47%, and 12%, respectively. The difference between *X. spinosum* and the plants without parasitization was significant, with a decrease of 22% and 25%, respectively (*p* < 0.05). The root biomass increased by 65% during parasitism during flowering, significantly higher than the increase under not parasitism (2%). The stem biomass increased by 26% and 14% during flowering and fruit parasitism, respectively, a significant difference from a decrease of 29% without parasitism (*p* < 0.05). The fruit biomass increased by 33% under parasitism during flowering and decreased by 38% under not parasitism, indicating a significant difference (*p* < 0.05). The increase in plant height during parasitism during the flowering and fruit stages was significantly higher than that of not parasitism. The total biomass and aboveground biomass of *X. italicum* parasitized during the seedling and flowering stages of *C. australis* decreased by 56%, 62%, and 54%, and 63%, significantly higher than the not parasitized reduction value (*p* < 0.05). The root biomass decreased by 61%, 54%, and 5% during the seedling, flowering, and fruit stages, respectively, with a significant difference from the 26% increase without parasitism (*p* < 0.05). The stem biomass decreased by 59%, 64%, and 6% during the seedling, flowering, and fruit stages, respectively, with a significant difference from an increase of 30% without parasitism (*p* < 0.05). The leaf biomass decreased by 54% and 59% during parasitism during the seedling and flowering stages, respectively, significantly higher than the decrease in not parasitism (6%) (*p* < 0.05). The parasitic reduction of fruit biomass by 60% during flowering was significantly higher than the not parasitic reduction value (12%) (*p* < 0.05). These results demonstrate that compared with that of the control, the parasitism of *C. australis* at different stages significantly increased the biomass of *X. spinosum* and significantly decreased the biomass of *X. italicum* (Figure 2).

Parasitism by *C. australis* significantly affected the biomass allocation of *X. spinosum* and *X. italicum*. Compared with that of the control, the root-to-shoot ratio of *X. spinosum* significantly increased during seedling parasitism; the leaf biomass ratio significantly increased during flowering parasitism; and the stem biomass ratio significantly increased during seedling, flowering, and fruit parasitism. The fruit biomass ratio significantly decreased during seedling, flowering, and fruit parasitism (*p* < 0.05). The root-to-shoot ratio and stem biomass ratio of *X. italicum* significantly increased during parasitism at the seedling, flowering, and fruit stages. The leaf biomass ratio significantly increased during parasitism at the seedling and flowering stages. The fruit biomass ratio decreased significantly during parasitism at the seedling, flowering, and fruit stages (*p* < 0.05). The results demonstrate that parasitism by *C. australis* caused *X. spinosum* and *X. italicum* to change their biomass allocation strategies, increasing the energy required for nutrient growth and decreasing the energy input for reproductive growth (Figure 3).

Under mixed planting conditions, *X. spinosum* parasitized *C. australis* during the flowering and fruit stages, with negative RCI values calculated based on both the individual aboveground biomass and total biomass, which were significantly less than zero (*p* < 0.05) under parasitized conditions during the flowering and fruit stages. The RCI of individual total biomass parasitized during the flowering and fruit stages was significantly lower than that under the not parasitized condition and seedling stages. The parasitism of *C. australis* during the flowering and fruit stages enhanced the competitive ability of *X. spinosum*. The RCI of *X. italicum*, calculated based on the root, aboveground, and total biomasses of a single plant, was positive and significantly greater than 0 (*p* < 0.05) during the parasitism of *C. australis* during the seedling and flowering stages, indicating that parasitism by *C. australis* during the seedling and flowering stages reduced the competitiveness of *X. italicum* (Table 2).

The parasitism of *C. australis* shortened the growth cycles of the two invasive weeds (Figure 4). Compared with the control, the growth period of *X. italicum* was shortened by 10 d, the fruiting period by 4 DAS, the growth period of *X. spinosum* by 9 DAS, and the fruiting period by 3 DAS. The growth periods of *X. italicum* and *X. spinosum* were significantly shortened, and the fruiting periods of both invasive weeds were significantly shortened.

Compared with that of the control, the parasitism of *C. australis* significantly advanced the budding time of *X. spinosum* and *X. italicum* by approximately 9 and 8 DAS, respectively. For flowering, the parasitism of *C. australis* significantly advanced the flowering time of *X. spinosum* by approximately 8 DAS and that of *X. italicum* by approximately 7 DAS. For fruiting, the parasitism of *C. australis* significantly advanced the fruiting time of *X. spinosum* by approximately 10 DAS and that of *X. italicum* by approximately 9 DAS. In addition, the parasitism of *C. australis* significantly advanced the fruit ripening time of *X. spinosum* and *X. italicum*, respectively, by approximately 13 and 12 DAS (Figure 5). These results demonstrate that the parasitism of *C. australis* not only shortened the growth cycle of the two invasive weeds, but also advanced them.

The parasitism of *C. australis* not only accelerated and shortened the life cycle of *X. spinosum* and *X. italicum,* but also changed the biomass and plant height indicators of the two invasive weeds. The total biomass, fruit biomass, plant height, and crown width of *X. spinosum* parasitized by *C. australis* decreased by 88%, 92%, 54%, and 70%, respectively; the total biomass, fruit biomass, plant height, and crown diameter of *X. italicum* decreased by 96%, 99%, 79%, and 71%, respectively, significantly different from those without parasitism (Figure 6). Thus, under the parasitism of *C. australis*, *X. spinosum* and *X. italicum* not only advanced their life cycle, but also inhibited their biomass, plant height, and crown width.

## 4. Discussion

*C. australis* had the greatest impact on plant growth and flowering when parasitized, increasing the biomass of *X. spinosum* by 48% and reducing the biomass of *X. italicum* by 62% compared with those of the not parasitized plants; it also increased the root-to-shoot ratio and decreased the fruit biomass ratio of *X. spinosum* and *X. italicum*, indicating that the parasitism of *C. australis* changed the growth and reproduction strategies of *X. spinosum* and *X. italicum*. By calculating the relative competitive intensity (RCI), the parasitism of *C. australis* enhanced the competitive ability of *X. spinosum*, and the competitive ability of *X. italicum* decreased. The parasitism of *C. australis* extended the growth period of *X. spinosum* by 9 d and the growth period of *X. italicum* by 10 d. This result indicates that although parasitism by *C. australis* enhanced the competitiveness of *X. spinosum*, it had an inhibitory effect on the growth of *X. spinosum* and *X. italicum*, which had a significant impact on the reproduction and diffusion of *X. spinosum* and *X. italicum* in invasive areas.

Often, biomass is an important parameter for measuring plant invasiveness, and a high biomass often leads to a strong reproductive ability and high fitness [28,29]. Additionally, plants are able to undergo morphological changes and respond to environmental changes by changing their competitive abilities above and below ground. Because the allocation of aboveground and underground parts affects the rate of resource acquisition, they have become an important feature of plant growth and competitive ability [30]. 

In the competition experiment, parasitism by *C. australis* increased the *X. spinosum* biomass and decreased the biomass of *X. italicum*. Moreover, *X. spinosum* and *X. italicum* had different resistances to *C. australis*: *X. spinosum* had a strong resistance and *X. italicum* had a relatively weak resistance. However, *X. spinosum* and *X. italicum* showed an increase in their root-to-shoot, stem, and leaf biomass ratios after the parasitism of *C. australis*, and their fruit biomass ratio decreased. 

Two hypotheses are related to environmental change and plant biomass allocation: the growth−defense trade-off hypothesis and the resource availability hypothesis. The growth−defense trade-off hypothesis suggests that there is a trade-off between plant growth and defense, that is, when plants are stressed by biological factors, such as temperature, water, and nutrients, or harmed by biological factors, such as herbivores and insects in adverse environments, they will reduce their investment in their growth and increase their investment in defense capabilities, such as increasing defense materials [31,32]. The resource availability hypothesis suggests that the relationship between natural enemies and resource availability can lead to changes in plant growth−defense trade-offs [33,34,35].

The changes in the biomass allocation of *X. spinosum* and *X. italicum* parasitized by *C. australis* demonstrated that *X. spinosum* and *X. italicum* reduced the biomass allocation of their fruits after being parasitized by *C. australis* and instead increased the biomass input of their roots, stems, and leaves. This is the result of invasive plants resisting natural enemies and environmental changes after entering new habitats. Therefore, the parasitic relationship between *C. australis* and *X. spinosum*, as well as *X. italicum*, supports the growth−defense trade-off and resource availability hypotheses. This investment trade-off benefits the survival, expansion, and rapid evolution of *X. spinosum* and *X. italicum*.

Competitive ability determines whether invasive plants successfully settle, reproduce, and spread after entering new habitats [36,37]. A strong competitive ability is conducive to resource competition between invasive plants, and a weak competitive ability is detrimental to invasive plants [38]. Moreover, views on the impact of parasitic plants on the competitive ability of invasive plants, such as improving competitive ability, reducing competitive ability, or maintaining stability, differ [39]. In the experiment on the impact of *C. australis* parasitism on the competitive ability of *X. spinosum* and *X. italicum*, the RCI of *X. spinosum* parasitized during the flowering and fruit stages was negative, and there was a significant difference from 0. Therefore, the parasitic relationship between *C. australis* and *X. spinosum* supports the evolutionary hypothesis of enhancing competitiveness. The parasitic relationship between *C. australis* and *X. italicum* during the seedling and flowering stages of *C. australis* also supports the evolutionary hypothesis of reduced competitiveness because the RCI calculated based on the total biomass per plant was positive and significantly different from 0.

Parasitism by *C. australis* affects the growth cycle of two invasive weeds of the genus Xanthium, to some extent, by absorbing nutrients and water. The results showed that *C. australis* parasitism significantly shortened the growth cycle of the two invasive weeds and significantly advanced the flowering time, fruiting time, fruit ripening time, biomass, plant height, and crown width. Individual height is an important physiological and ecological characteristic of plants and is closely related to their life history [40]. Plant height is an important trait that measures plant function and reflects the nutrient balance between plant growth and reproduction, affecting plant phenology [41]. Studies have shown [42] that increasing plant height delays its flowering phenology, and plants with shorter plant heights enter the reproductive period earlier to avoid competition with taller plants for light [43,44]. This is consistent with the following: *C. australis* parasitism in this experiment resulted in a decrease in the plant height of the two invasive weeds and an earlier flowering time for the two invasive weeds.

## 5. Conclusions

Compared with the control, parasitism by *C. australis* increased the biomass of *X. spinosum*, and the biomass of *X. italicum* decreased. This indicates that *X. spinosum* has higher resistance to *C. australis* than *X. italicum* does. Under parasitism by *C. australis*, the root-to-shoot, stem, and leaf biomass ratios of *X. spinosum* and *X. italicum* increased, and the fruit biomass ratio decreased, indicating that *C. australis* parasitism changed the biomass distribution of *X. spinosum* and *X. italicum*. When *C. australis* parasitized during the flowering and fruit stages, the RCI of *X. spinosum*, calculated based on the total biomass of a single plant, was negative, indicating that *C. australis* parasitized to enhance the competitive ability of *X. spinosum*. When *C. australis* parasitized during the seedling and flowering stages, the RCI of Italian *X. italicum* calculated based on the total biomass per plant was positive, indicating that *C. australis* parasitized and reduced the competitiveness of Italian *X. italicum*. When experiencing the parasitism of *C. australis*, the growth cycle of *X. spinosum* and *X. italicum* was shortened, especially in the fruit stage. Additionally, the parasitism of *C. australis* significantly advanced the flowering time, fruiting time, and fruit ripening time of *X. spinosum* and *X. italicum*. The parasitism of *C. australis* significantly reduced the biomass, plant height, and crown width of *X. spinosum* and *X. italicum*. Thus, the parasitic effect of *C. australis* on the growth cycle of *X. spinosum* and *X. italicum* was mainly manifested by shortening and advancing the growth cycle of *X. spinosum* and *X. italicum* and reducing their biomass and other growth indicators.

## Figures and Tables

**Figure 1 biology-13-00023-f001:**
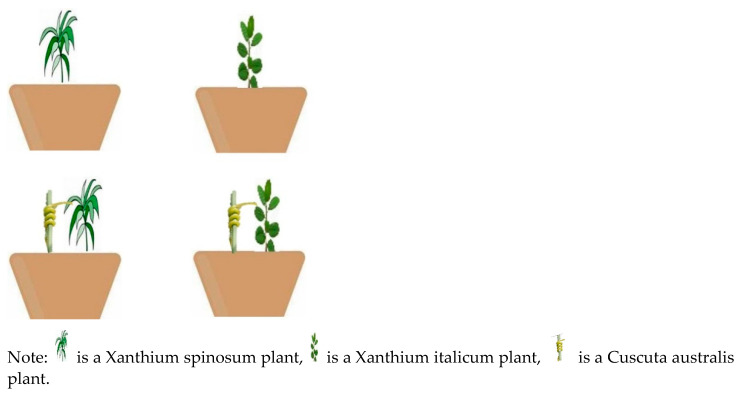
Treatment of the phenological effects of *Cuscuta australis* parasitism on two invasive weeds.

**Figure 2 biology-13-00023-f002:**
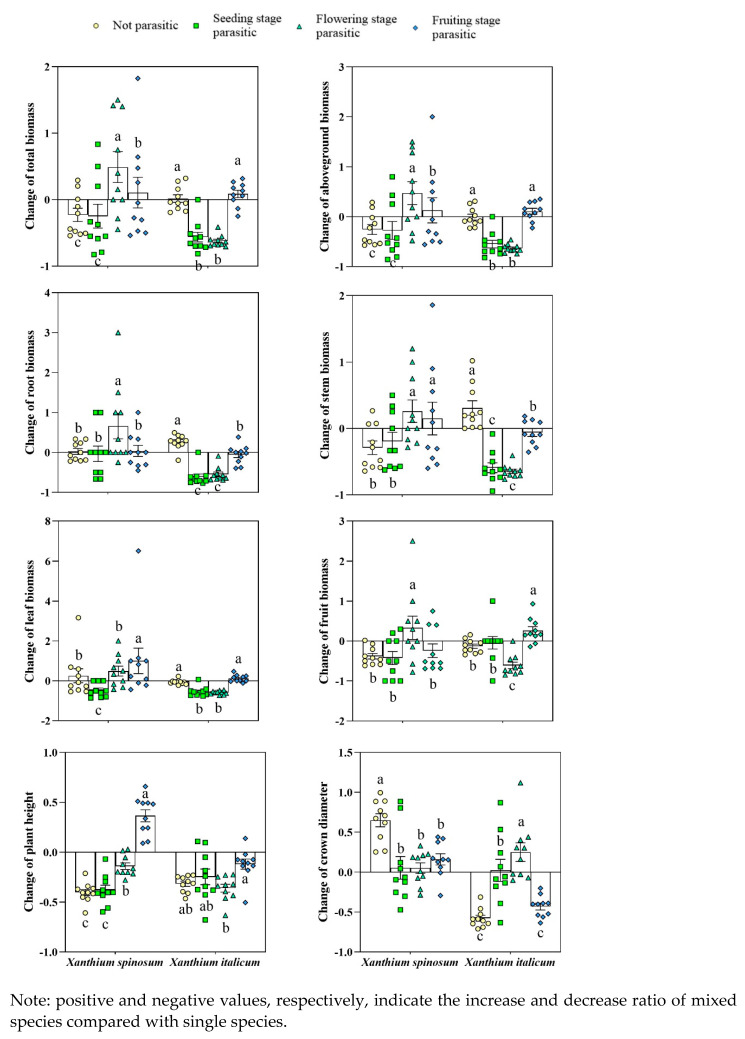
Biomass changes in *Xanthium spinosum* and *Xanthium italicum* parasitized by *Cuscuta*. Different lowercase letters indicate differences in the stages of between *Xanthium spinosum* and *Xanthium italicum* by parasitism *Cuscuta*.

**Figure 3 biology-13-00023-f003:**
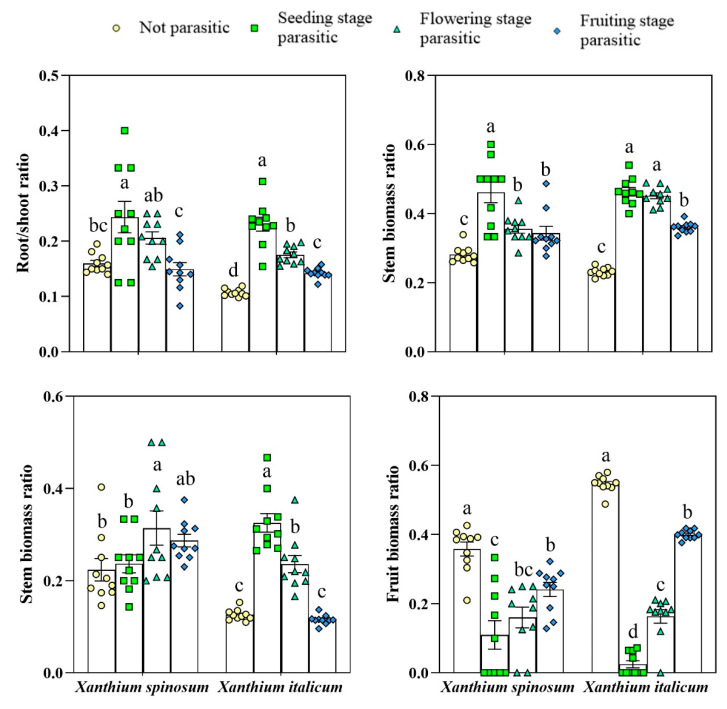
Biomass distribution of *Xanthium spinosum* and *Xanthium italicum* parasitized by *Cuscuta australis*. Different lowercase letters indicate differences in the stages of between *Xanthium spinosum* and *Xanthium italicum* by parasitism *Cuscuta*.

**Figure 4 biology-13-00023-f004:**
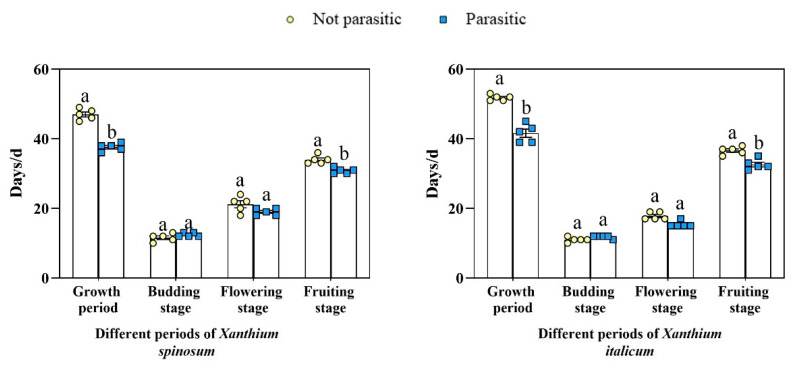
Comparison of number of days in different phenological periods of *Xanthium spinosum* and *Xanthium italicum* parasitized by *Cuscuta australis*. Different lowercase letters indicate differences in the stages of between *Xanthium spinosum* and *Xanthium italicum* by parasitism *Cuscuta*.

**Figure 5 biology-13-00023-f005:**
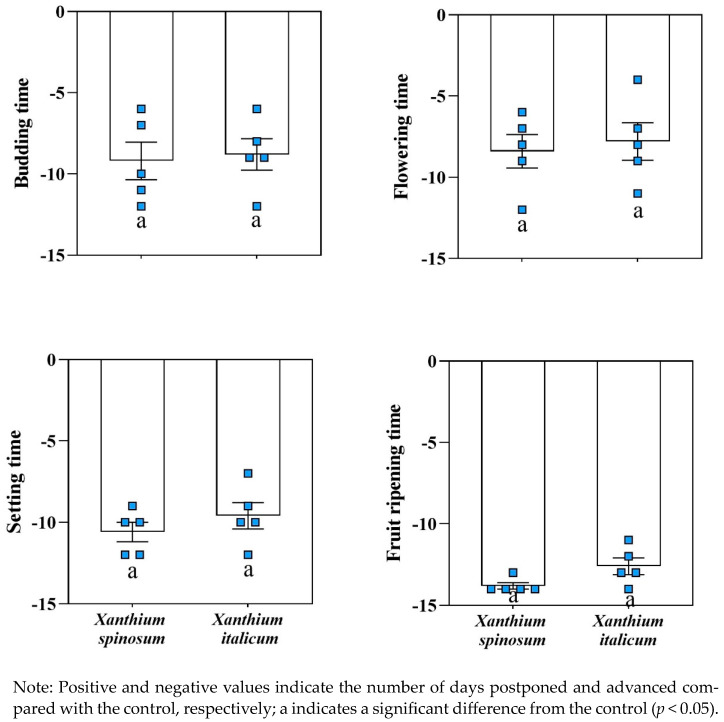
Temporal variation in *Xanthium spinosum* and *Xanthium italicum* parasitized by *Cuscuta australis*.

**Figure 6 biology-13-00023-f006:**
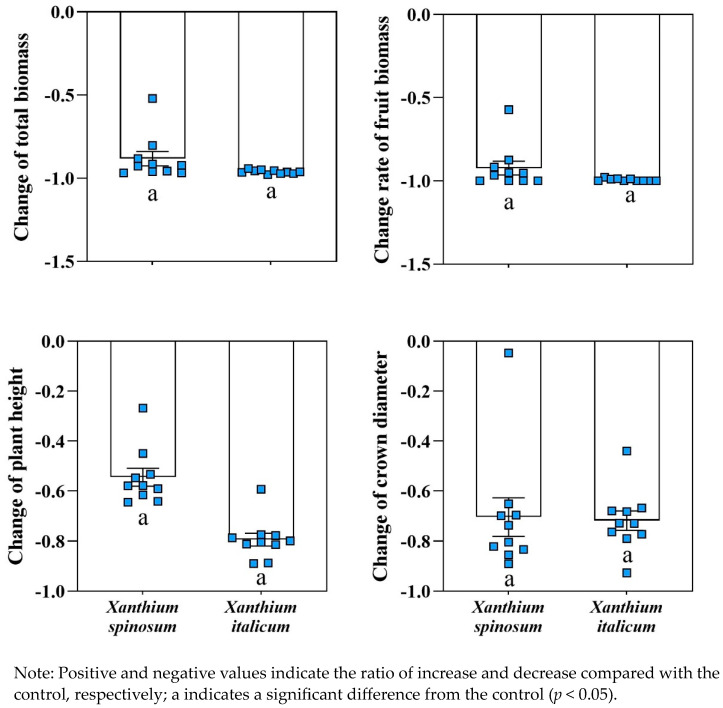
Changes in total biomass, fruit biomass, plant height, and crown amplitude of *Cuscuta australis* parasitism on *Xanthium spinosum* and *Xanthium italicum*.

**Table 1 biology-13-00023-t001:** Seed collection sites of experimental materials.

Species	Locality	Geographical Coordinates	Altitude (m)	Habitat
*Cuscuta australis*	Sanping Farm in Urumqi	43°56′ N, 87°20′ E	790	Farmland
*Xanthium spinosum*	Sanping Farm in Urumqi	43°56′ N, 87°20′ E	790	Forest belt
*Xanthium italicum*	Sanping Farm in Urumqi	43°56′ N, 87°20′ E	790	Forest belt

**Table 2 biology-13-00023-t002:** Relative competition intensity of *Xanthium spinosum* and *Xanthium italicum* parasitized by *Cuscuta australis* (Mean ± SE).

Treatment	*Xanthium spinosum*			*Xanthium italicum*		
	Root Biomass	Aboveground Biomass	Total Biomass	Root Biomass	Aboveground Biomass	Total Biomass
Not parasitic	−0.023 ± 0.021 b	0.255 ± 0.03 a	0.229 ± 0.029 a	−0.261 ± 0.018 c	0.009 ± 0.018 b	−0.016 ± 0.001 b
Seeding stage parasitic	−0.083 ± 0.04 b	0.277 ± 0.053 a	0.248 ± 0.053 a	0.612 ± 0.021 a,*	0.549 ± 0.022 a,*	0.563 ± 0.021 a,*
Flowering stage parasitic	−0.15 ± 0.058 b	−0.472 ± 0.06 b,*	−0.489 ± 0.07 b,*	0.549 ± 0.174 a,*	0.635 ± 0.08 a,*	0.622 ± 0.082 a,*
Fruiting stage parasitic	0.681 ± 0.15 a,*	−0.127 ± 0.075 ab	−0.106 ± 0.07 b	0.056 ± 0.002 b	−0.107 ± 0.017 b	−0.086 ± 0.006 b

Note: Different letters in the same column indicate significant differences between treatments (*p* < 0.05), * indicates significant differences between RCI and 0 (*p* < 0.05).

## Data Availability

The original contributions presented in the study are included in the article; further inquiries can be directed to the corresponding author.

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
