# Peer review of "Effect of Parasitic Native Plant *Cuscuta australis* on Growth and Competitive Ability of Two Invasive Xanthium Plants"

_biology, 2023, doi:10.3390/biology13010023_

Round 1
Reviewer 1 Report
Comments and Suggestions for Authors
According to the POWO database (https://powo.science.kew.org/), Xanthium italicum Moretti is a synonym of Xanthium orientale L.
If the authors took 3 fruits from a plant, they got 3 seeds per plant, because Xanthium species have only one seed per fruit. Why was it necessary to mix three seeds (116)? How did the authors remove underdeveloped seeds (118) if there is only one seed in the fruit?
How many seeds were in the study? Did I understand correctly that there were 150?
The conclusion (341-342) ... "C. australis parasitism in this experiment resulted in a decrease in the plant height of the two invasive weeds" somewhat contradicts the conclusion (346) "... parasitism by C. australis increased the biomass of X. spinosum, and the biomass of X. italicum decreased". Why is the biomass of X. spinosum increasing? A very surprising conclusion.
Comments on the Quality of English LanguageThe experimental materials were C. australis, X. spinosum, and X. italicum (111) - there is no such word order in the English language
Reviewer 2 Report
Comments and Suggestions for Authors
The manuscript 'Effect of parasitic native plant Cuscuta australis on growth and competitive ability of two invasive Xanthium plants" by He et al. is exciting research. The study is novel and provides some intriguing new hypotheses for studying plant-plant relationships. Their research highlights parasite plants' possible influence on plant communities' structure and offers a spotlight on the intricate relationships between native and invasive species. This study delivers insightful information for ecological management plans and creates new vistas for investigating the causes behind plant invasions. Future studies in this field can expand on the groundwork established by this fascinating study and advance our understanding of plant ecology and management. I have a few recommendations to make them even better.
Minor comments:
1. Please change the type of article from "Essay" to "Article".
2. Line 100: "The aim is....."Please write it in the past tense.
3. Line 171: Please mention the accuracy of the balance used.
4. The authors use the "d" character for days, confusing for a single word, so please replace it with days throughout the paper. For 5 days after sowing (Line 141 as well as similar), it is acceptable to write "5 days after sowing (DAS)" when it appears for the first time. And then, throughout the paper, you can write it as, for example, 8 DAS, 9 DAS, etc.
5. Line 182, "C australis" full stop is missing after C.
6. The conclusion should be in a single paragraph.
7. Line 182-187:The total stem parasitic plant C. australis takes food, minerals, and water from the host plant. According to your findings, the parasitic plants impound on two different modes on both host plants (X. spinosum and X. italicum), i.e., in one plant, it has a positive impact on biomass and fruit, while in another plant, it has a negative impact. Firstly, please recheck your data to confirm the results. Secondly, what is the exact physiological mechanism of these contradictory results?
8. Referencing is not according to journal guidelines. Please follow the journal format for references.
Round 2
Reviewer 1 Report
Comments and Suggestions for Authors
From the answers, it was clear that the authors did not mix seeds from a SAME plant, but from ALL plants. It should be written as follows (126 line): seeds collected from all X. spinosum plants were mixed. Similarly, the seeds collected from all selected Xanthium italicum plants were mixed.
Comments on the Quality of English LanguageThis text should be edited by a professional translator
Author Response
Comments 1: [From the answers, it was clear that the authors did not mix seeds from a SAME plant, but from ALL plants. It should be written as follows (126 line): seeds collected from all X. spinosum plants were mixed. Similarly, the seeds collected from all selected Xanthium italicum plants were mixed.]
Response 1: Thank you for pointing this out. We agree with this comment. Therefore,we have modified this to Seeds collected from all X. spinosum plants were mixed. Similarly, the seeds collected from all selected X. italicum plants were mixed. (Line 126 to line 128)